# Aberrant auditory prediction patterns robustly characterize tinnitus

**Lisa Reisinger[1]\*[†], Gianpaolo Demarchi[1†], Jonas Obleser[2,3], William Sedley[4], Marta Partyka[1], Juliane Schubert[1], Quirin Gehmacher[5], Sebastian Roesch[6], Nina Suess[1], Eugen Trinka[1,7,8], Winfried Schlee[9], Nathan Weisz[1,8]**

[1]Centre for Cognitive Neuroscience and Department of Psychology, Paris-Lodron-University of Salzburg, Salzburg, Austria; [2]Department of Psychology, University of Lübeck, Lübeck, Germany; [3]Center of Brain, Behavior and Metabolism, University of Lübeck, Lübeck, Germany; [4]Translational and Clinical Research Institute, Newcastle University, Newcastle upon Tyne, United Kingdom; [5]Wellcome Centre for Human Neuroimaging, University College London, London, United Kingdom; [6]Department of Otolaryngology, University Hospital Regensburg, Regensburg, Germany; [7]Department of Neurology, Christian Doppler University Hospital, Paracelsus Medical University, Salzburg, Austria; [8]Neuroscience Institute, Christian Doppler University Hospital, Paracelsus Medical University, Salzburg, Austria; [9]Department of Psychiatry and Psychotherapy, University of Regensburg, Regensburg, Germany

**\*For correspondence:**
lisa.reisinger@plus.ac.at

[†]These authors contributed equally to this work

## eLife Assessment

This **important** work presents two studies on predictive processing in subjects with and without tinnitus, matched for age, sex and hearing loss. These studies together provide **compelling** evidence for an enhanced predictability of upcoming sounds in regular sequences in EEG data recorded from tinnitus subjects. This work will be of interest to researchers, especially neuroscientists, in the tinnitus field and beyond.

**Abstract** Phantom perceptions like tinnitus occur without any identifiable environmental or bodily source. The mechanisms and key drivers behind tinnitus are poorly understood. The dominant framework, suggesting that tinnitus results from neural hyperactivity in the auditory pathway following hearing damage, has been difficult to investigate in humans and has reached explanatory limits. As a result, researchers have tried to explain perceptual and potential neural aberrations in tinnitus within a more parsimonious predictive-coding framework. In two independent magnetoencephalography studies, participants passively listened to sequences of pure tones with varying levels of regularity (i.e. predictability) ranging from random to ordered. Aside from being a replication of the first study, the pre-registered second study, including 80 participants, ensured rigorous matching of hearing status, as well as age, sex, and hearing loss, between individuals with and without tinnitus. Despite some changes in the details of the paradigm, both studies equivalently reveal a group difference in neural representation, based on multivariate pattern analysis, of upcoming stimuli before their onset. These data strongly suggest that individuals with tinnitus engage anticipatory auditory predictions differently to controls. While the observation of different predictive processes is robust and replicable, the precise neurocognitive mechanism underlying it calls for further, ideally longitudinal, studies to establish its role as a potential contributor to, and/or consequence of, tinnitus.

## Introduction

Subjective perceptual awareness is based on huge amounts of environmental inputs (i.e. sensations), which are transduced by sensory receptors. Phantom perceptions are peculiar in that they cannot be explained by sensory input. In the case of tinnitus, individuals consciously perceive one or more pure tones or narrowband noises that lack any identifiable source in the environment or the body (*Baguley et al., 2013*).

Approximately 10–15% of the young to middle-aged adult population experience tinnitus as a common auditory phantom perception, with greater prevalence of 24% in older adults (*Biswas et al., 2022*; *Henry et al., 2020*; *Jarach et al., 2022*). For a smaller portion of the population, the sensation of bothersome tinnitus poses a significant detriment to quality of life, due to reduced sleep quality, substantially increased distress, and anxiety (*Dobie, 2003*) – all largely independent of the intensity or duration of the phantom perception (*Kandeepan et al., 2019*; *Meyer et al., 2014*).

Which neural mechanisms contribute to the generation and maintenance of chronic tinnitus remains unresolved. Hearing loss has been identified as a main risk factor for tinnitus (*Kim et al., 2015*). Indeed, for 75–80% of people with tinnitus, objective audiometric testing indicates hearing loss (*Wallhäusser-Franke et al., 2017*). Previous findings support the idea that some form of auditory damage – even without clear audiometric changes – facilitates tinnitus development (*Roberts et al., 2006*; *Schaette and McAlpine, 2011*; *Schaette et al., 2012*; *Weisz et al., 2006*) and provokes maladaptive changes.

Based on the observation of increased neural activity following hearing loss in animal models (*Eggermont and Roberts, 2004*; *Roberts et al., 2010*), a still-influential 'altered-gain' view holds that reduced auditory input following hearing damage leads neurons in the auditory pathway to increase their responsivity, thereby restoring their mean activity level; in this framework, the perception of phantom sounds is a 'downside' to this homeostatic process, as spontaneous activity can engage downstream auditory regions (*Schaette and Kempter, 2006*; *Sedley, 2019a*). Other influential frameworks in tinnitus are the thalamocortical dysrhythmia theory (*Brinkmann et al., 2021*; *Llinás et al., 1999*), the noise cancellation model (*Rauschecker et al., 2010*; *Song et al., 2015*) as well as theories about map reorganization (*Eggermont and Roberts, 2012*) or approaches regarding specific neural networks in tinnitus (*De Ridder et al., 2011*; *Schlee et al., 2009*).

However, apart from a significant shortage of data bridging animal and human research in these different frameworks, empirical support in humans is weak, difficult to replicate, and marked by strong interindividual variability (*Eggermont and Roberts, 2015*; *Elgoyhen et al., 2015*; *Reisinger et al., 2023*). Beyond the lack of solid evidence, the models face further explanatory challenges (*Sedley, 2019a*): (1) People with hearing loss do not necessarily experience tinnitus (*Wallhäusser-Franke et al., 2017*). (2) The onsets of tinnitus and hearing loss often do not occur at the same time (*Roberts et al., 2010*). (3) Not all cases of acute tinnitus transform into chronic tinnitus (*Mühlmeier et al., 2016*; *Vielsmeier et al., 2020*). Overall, this situation calls for the pursuit of alternative or complementary models that place less emphasis on the hearing status of the individual.

One attempt along these lines has been the development of a Bayesian inference framework for tinnitus perception (*Sedley et al., 2016a*). This framework emphasizes the constructive nature of perception being guided by internal models (*Helmholtz, 1867*). Therein, sensory input is dynamically compared to predictions or so-called priors. The framework holds that spontaneous activity in the auditory pathway acts as a precursor of tinnitus. In the healthy auditory system, spontaneous activity is 'ignored', due to the default prior of silence. However, certain circumstances can shift this prior, such that a particular sound is expected (*Hullfish et al., 2019*; *Sedley et al., 2016b*). This conceptual model bridges several explanatory gaps: for example, the inconsistent findings in humans regarding the 'altered gain' view which states altered neural activity in the auditory pathway. Recent findings vary in both the targeted frequency bands and the direction of the reported power changes which impede consistent conclusions (*Eggermont and Roberts, 2015*; *Elgoyhen et al., 2015*; *Reisinger et al., 2023*). The Bayesian inference framework could, therefore, explain the experience of tinnitus in lieu of any increase in neural activity in the auditory system, or indicate an additional alteration, on top of hearing loss, for tinnitus to be perceived.

However, convincing empirical support is still sparse, due to the difficulty of deriving robust measures for tinnitus-supporting priors from ongoing brain activity. Few studies have provided support for altered prediction processes in tinnitus, which is in line with the predictive-coding framework using

either EEG evoked responses (*Mohan et al., 2022*; *Sedley et al., 2019b*) or computational modeling (*Hu et al., 2021*). Furthermore, the question of why only some individuals would shift their priors, thus developing tinnitus, remains unclear.

In this work we hypothesized that, given that forming a strong sensory prediction based on weak sensory evidence is thought to be a key process in developing tinnitus, individuals with stronger predictive tendencies are more vulnerable to developing tinnitus (this has similarities to the strong prior hypothesis of hallucinations developed by *Corlett et al., 2019*). We utilized a powerful, recently established experimental approach (*Demarchi et al., 2019*) showing anticipatory activations of tonotopically specific auditory templates for regular tone sequences. This method allows us to explicitly investigate predictive patterns in line with the Bayesian inference framework (*Sedley et al., 2016b*), leading toward the overall question whether alterations in predictive coding can be interpreted as a neural correlate of tinnitus or rather as a risk factor. Since this question can solely be targeted in a longitudinal manner, we aimed in a first step to investigate prediction patterns in tinnitus over two independent samples, deriving robust effects that should be considered in future research.

In a first sample we identified evidence for aberrant predictive processing in tinnitus. However, audiometric data was not consistently assessed for the participants without tinnitus. Therefore, conclusions that our identified patterns are related to tinnitus rather than hearing loss could not be drawn with certainty. To control for this possible confound, we recruited a large, new sample in which individuals with and without tinnitus were matched for hearing loss. By employing a highly similar experimental design—albeit with some minor modifications, such as a reduced number of trials and narrower carrier-frequency range—and identical analysis methods, we replicated our previous key findings, thereby reinforcing the initial claims. This second study was by itself successfully accepted as a registered report (https://osf.io/8bv29, *Chambers, 2024*). Altogether, over two experiments, our study presents robust and replicable evidence for aberrant predictive processing in tinnitus that cannot be explained by hearing loss.

## Methods
### Participants

In Study 1, a total of 34 individuals with tinnitus (17 females, age 20–67 years, mean = 45.12, sd = 13.65) participated in the experiment. Within this group, 25 individuals (16 females, age 20–66 years, mean = 40.92, sd = 13.17) were age-matched (in all cases but one both age- and sex-matched) with the control group and used for group comparisons. Laterality of the participants was not assessed. Tinnitus-related questionnaires (German version of *Tinnitus Questionnaire*, TQ (*Goebel and Hiller, 1992*), *Tinnitus Sample Case History Questionnaire*, TSCHQ (*Langguth, 2011*), and 10-point scale *Tinnitus Severity*, TS) were collected for individuals with tinnitus. Standardized pure-tone audiometric testing for frequencies from 125 Hz to 8 kHz was performed in 31 out of 34 tinnitus participants using Interacoustic AS608 audiometer. Averages were computed over all frequencies and both ears. 25 individuals (16 females, age 21–65 years, mean = 41.56, sd = 13.68) reporting no relevant audiological, neurological or psychiatric treatment history took part as a control group. Control subjects were age-matched to each tinnitus participant by the ±3 years criterion, selecting the closest match in cases where more than one subject was eligible. No differences were shown for age between the samples (*t* = 0.17, p = 0.89). As pure-tone audiometric testing was not included for the control subjects, group comparisons between hearing thresholds were not feasible. All participants provided written informed consent prior to participating.

In Study 2, 40 individuals with tinnitus (16 females, age 24–74 years, mean = 57.73, sd = 14.12), as well as 40 hearing-, age-, and sex-matched control subjects (16 females, age 24–76 years, mean = 57.43, sd = 13.94) completed the experiment. For the tinnitus group, inclusion criteria were a tinnitus duration of more than 6 months. No participants with psychiatric or neurological diseases were included in the sample. Laterality of the participants was not assessed. We provided questionnaires covering tinnitus (German short version of Tinnitus Questionnaire, Mini-TQ *Goebel and Hiller, 1992*) and hearing characteristics (German version of the Speech, Spatial and Qualities of Hearing Scale, SSQ *Kiessling et al., 2011*), along with an online hearing test (Shoebox, Ottawa, Canada). The Mini-TQ includes subscales targeting emotional distress, cognitive distress, and sleep disturbances which we will use to draw conclusions about the impact of tinnitus distress (*Hiller and Goebel, 2004*).

We also performed pure-tone audiometric testing and defined hearing loss by a hearing threshold higher than 30 dB HL in at least one frequency. Four individuals with tinnitus did not show any audiometric abnormality; four of the participants showed unilateral hearing impairments on at least one frequency; 26 volunteers had high-frequency hearing loss (i.e. hearing thresholds higher than 30 dB); and six individuals were hearing impaired over most frequencies (i.e. hearing thresholds higher than 30 dB). The control group was recruited afterwards, in order to match the distribution of the tinnitus group by age, sex, and hearing status. Accordingly, we aimed to find the best possible match that our data allowed for between individuals with tinnitus and control subjects regarding the results of the audiometry. We calculated the individual mean hearing ability based on the values for 500, 1000, 2000, and 4000 Hz, which is a common approach for averaging results of pure-tone audiometry (i.e. PTA-4, see e.g. *Lin et al., 2011*; *Ozdek et al., 2010*). Using independent *t*-tests, we found no differences in hearing status over frequencies between groups for the left ($t = -1.19$, p = 0.238) and right ear ($t = -1.72$, p = 0.09). An additional linear regression including all frequencies from 125 to 8000 Hz also showed that hearing thresholds did not differ between ears ($b = 0.311$, SE = 1.600, p = 0.846) and groups ($b = 1.702$, SE = 1.553, p = 0.273), but solely between frequencies ($b = 0.003$, SE = 0.000, p < 0.001). Interactions were not significant as well. Control subjects were age-matched to each tinnitus participant by a ±2-year criterion, choosing the closest match when more than one subject was suitable. All participants provided written informed consent before participating. The experimental protocols were approved by the ethics committee of the University of Salzburg (EK-GZ: 22/2016 with Addenda).

## Stimuli and experimental procedure

Prior to entering the shielded magnetoencephalography (MEG) room, we applied five head position indicator (HPI) coils to the scalp of each participant. We used a Polhemus FASTRAK (Polhemus, Colchester, Vermont, USA) digitizer to register head shape and position for each individual by marking nasion and left/right pre-auricular points, location of the HPI coils and approximately 300 additional points over the scalp. After this preparation, we performed a 5-min resting-state recording (not included in the analyses). Next, participants passively listened to sound sequences without further instruction, while watching a silent movie ('Cirque du Soleil: Worlds Away' in Study 1 and a nature documentary in Study 2). The movie was displayed using a projector (PROPIXX, VPixx Technologies, Canada) and a periscope onto a screen inside the shielded room. Auditory stimulation was presented to both ears via MEG-compatible pneumatic in-ear headphones (SOUNDPixx, ibid).

In Study 1, four different pure (sinusoidal) tones were presented, with carrier frequencies logarithmically spaced between 200 and 2000 Hz (200, 431, 928, and 2000 Hz) analogous to the original paradigm (*Demarchi et al., 2019*). In Study 2, carrier frequencies of the four tones were logarithmically spaced between approximately 440 and 1043 Hz (i.e. 440, 587, 782, and 1043 Hz). We reduced the carrier frequencies to a maximum of 1043 Hz to ensure that the sounds provided were within a region of normal audiometric thresholds. Specifically, we aimed to avoid potential effects of high-frequency hearing loss on the highest frequency tones. In both designs, each tone lasted 100 ms, tapered at both ends with 5 ms linearly ascending/descending periods, and we presented the sounds at a constant 3-Hz stimulation rate. Sound intensity was individually determined by presenting a short audio sequence to the participants and adjusting the loudness according to an individual pleasant volume with all four frequencies audible for the participant.

In Study 1, we presented four blocks of tone sequences comprising 4000 stimuli, each lasting approximately 22 min. The number of particular tone frequencies was balanced across blocks, so the condition-blocks varied solely by presentation order, which was parametrically modulated in their regularity (entropy) level using different transition matrices (*Nastase et al., 2014*). We used four entropy conditions for the design. The random condition had the highest entropy (i.e. the lowest regularity), and the transition probabilities from one sound to another were equal, preventing any possibility of accurately predicting upcoming stimuli. In contrast, the ordered condition had the lowest entropy level (i.e. the highest regularity), and in 75% of trials, one sound was followed by a specific other sound. In 25% of trials, the same sound was repeated. Additionally, two intermediate entropy conditions were included, labeled here as midminus and midplus. To control for the influence of self-repetitions, the diagonal of the transition matrices was set to be always 25% across all entropy conditions (see *Figure 1*). Furthermore, a randomly chosen 10% of the sounds were omitted in all

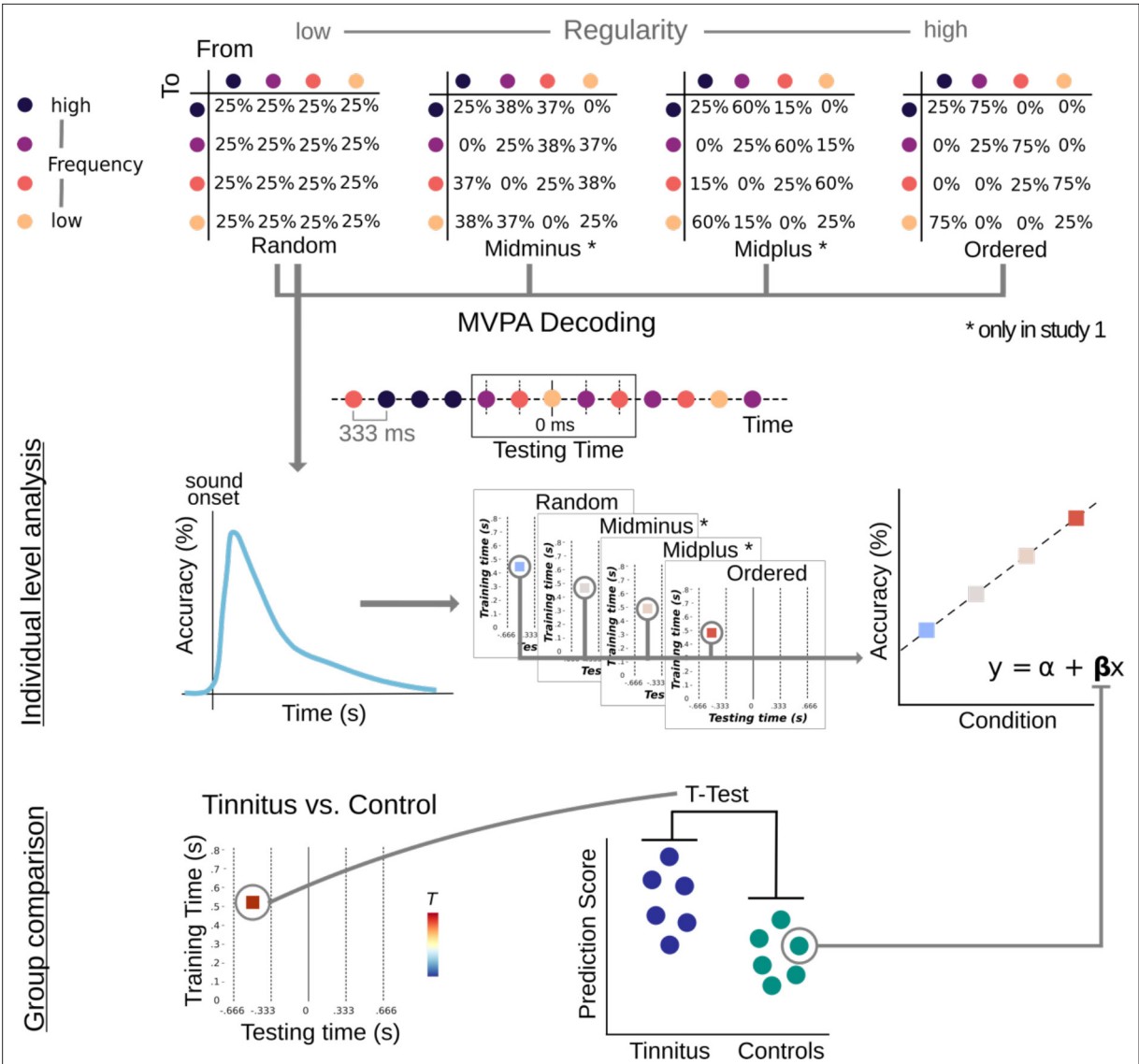

**Figure 1.** Experimental design and analysis rationale. **Upper panel:** Transition matrices used to generate sound sequences according to the different conditions (random, midminus, midplus, and ordered) with a schematic example of a brief sound sequence. The 'Testing Time' window corresponds to one trial with the to-be-decoded carrier frequency in the center (at 0 ms; marked by solid line), preceded and followed by two other tones (marked by dashed lines). **Middle panel:** Individual-level analysis. For multivariate pattern analysis (MVPA), time-shifted classifiers were trained on events in the random condition (left panel) and applied in a condition- and time-generalized manner to all conditions (middle panel). **Lower panel:** Group comparisons. At a group level, the resulting slopes ($\beta$-coefficients) of the regression analysis were compared between the tinnitus group and the control group. Notably, Study 2 solely included random and ordered sound conditions, as well as a narrower frequency range between ~400 and 1000 Hz. Analyses approaches did not vary except statistics were based on differences in decoding accuracy between conditions (referred to as 'neural predictions scores') rather than $\beta$-coefficients.

conditions (not included in the current analysis), resulting in 900 sound trials and 100 omission trials per entropy block, that is 3600 sounds and 400 omissions in total.

In Study 2, fewer stimuli were used, and we combined the sound sequences solely into two continuous blocks, each lasting approximately 8 min. Furthermore, no omissions were included, and participants solely listened to sounds that were either random or ordered. We balanced the number of stimuli across blocks, and each block contained 1500 particular tone frequencies with groups of 500 consecutive stimuli following the same regularity level. To balance the number of conditions, one of the two blocks started with a random condition (500 stimuli), followed by an ordered sequence (500 stimuli) and ended with a random condition (500 stimuli). For the other block, sounds started

accordingly in an ordered condition, followed by random sounds and a second sequence of ordered sounds. Therefore, data collection comprised 1500 stimuli of each condition. Both experiments were written using the MATLAB-based (version 9.1 The MathWorks, Natick, Massachusetts, USA) Psychophysics Toolbox (*Brainard, 1997*).

## MEG data acquisition and preprocessing

We measured magnetic brain activity using a whole-head MEG (Triux, MEGIN Oy, Finland), in which brain signals were captured by 102 magnetometers and 204 orthogonally placed planar gradiometers. Participants sat in a dimly lit magnetically shielded room (AK3b, Vacuumschmelze, Germany) and were measured with a sampling rate of 1000 Hz and default hardware filters set by the manufacturer (0.1 Hz high pass to 330 Hz low pass). We used a signal-space separation algorithm (SSS *Taulu and Kajola, 2005*) implemented in the Maxfilter program (version 2.2.15) to reduce external noise from the MEG signal (mainly 16.6 Hz, and 50 Hz-plus harmonics) and to realign data of different measurement blocks to a common standard-head position ('-trans default' Maxfilter parameter), based on the head position measured at the beginning of each block (*Cichy and Pantazis, 2017*). Additionally, the Maxfilter algorithm detected bad channels, removed and interpolated the data.

The analyses were based on magnetometers only, since information between magnetometers and gradiometers is mixed after the Maxfilter step (*Garcés et al., 2017*) and were carried out with our own scripts, including the Fieldtrip toolbox (*Oostenveld et al., 2011*). For preprocessing the data, we applied a high-pass filter at 0.1 Hz (sixth-order zero-phase Butterworth filter), as well as a low-pass filter at 30 Hz, to the raw data and used it as an input for an Independent Component Analysis (ICA) algorithm. Next, we inspected the ICA components visually to detect and remove unwanted artifacts, such as eye blinks and movements, heartbeats and 16⅔ Hz artifacts (the frequency of Austrian train power supply). In Study 1, an average of 3.6 components (sd = 1.2) were removed. In Study 2, an average of 2.3 components (sd = 0.72) were removed in the tinnitus group and an average of 2.25 components (sd = 0.67) in the control group. After eliminating these components, we epoched the continuous data of Study 1 into chunks from 1000 ms before to 1000 ms after sound/omission onset. The data of Study 2 was epoched from 400 ms before to 500 ms after sound onset. This approach enabled analysis of both regularity-dependent pre-activations and post-stimulus decoding accuracies. In a final step, we downsampled the data to 100 Hz to further use it for multivariate pattern analyses (MVPAs).

## MVPA and classifier weights projection

We used MVPA as implemented in the MVPA-Light toolbox (https://github.com/treder/MVPA-Light; *Treder, 2024*), which was modified to extract classifier weights. We defined four target classes in line with the frequencies of the sound presented in each specific trial. To avoid potential carryover effects from previous sounds and to focus exclusively on carrier-frequency-related information and the corresponding neural templates, we trained the classifier solely on the random sound sequences.

We trained a multiclass linear discriminant analysis classifier on each sample point of the random condition and averaged the classification accuracy for each subject at a group level for further comparisons. Additionally, we used a temporal generalization method (*King and Dehaene, 2014*) to analyze the ability of the classifier to generalize across time points in the training set to time points in the testing set. When testing on the ordered condition, we did not perform any cross-validation, as our approach already consisted of cross-decoding. For testing on the random tones, we performed a fivefold cross-validation. It is further important to specify that we trained on the post-stimulus interval and tested on the pre-stimulus interval of the random tones.

In a final step, we extracted the training decoder weights of relevant pre-stimulus time frames and projected them in the source space in order to localize the informative activity of carrier-frequency processing (*Demarchi et al., 2019*; *Marti and Dehaene, 2017*). We computed single-shell head models (*Nolte, 2003*) by co-registering the head shapes of the participants with a standard brain template from the Montreal Neurological Institute (MNI, Montreal, Canada). A grid with 1-cm resolution and 2982 voxels was morphed to fit the individual brain volumes of the participants. As a result, we were able to perform group-level comparisons, since all grid points belong to the same brain regions across subjects.

## Statistical analysis

For the analysis of Study 1, we compared decoding accuracy in the random condition against chance level (25%) using a *t*-test at every individual time point from −0.2 to 1 s. To correct for multiple comparisons, the p-value of 0.05 was Bonferroni corrected over time points. We arranged time-generalized accuracy results for sounds from random to ordered in both groups and computed a regression for each single point of the testing–training accuracy matrices, using the MATLAB built in least square *mldivide* algorithm ('\'). This resulted in a training time by testing time matrix of slopes ('*β*') for each subject, discarding intercepts. To compare the groups (25 tinnitus subjects vs 25 age-matched controls), we ran a *t*-test between the two matrices with coefficients obtained in the regression step, inputting them in the form of time–frequency 2D structures (time-generalized *β* values) in the ft_freqstatistics Fieldtrip function. For each individual, we quantified the modulation of decoding accuracy by regularity condition as the slope (*β*-coefficient) of a linear regression analysis (see *Figure 1*). In order to account for multiple comparisons, we used a nonparametric cluster permutation test (*Maris and Oostenveld, 2007*), with 1000 permutations and a p < 0.05 to threshold the clusters. We pursued further analysis with questionnaire data using R (v4.1.2; *R Development Core Team, 2020*). In the whole sample of participants with tinnitus (*n* = 34) we performed a Spearman correlation of the *β*-coefficient values corresponding to the time point of the maximum and the minimum *t*-value in intergroup analysis (comprised of positive and negative significant clusters emerging in group comparison for sound trials) with hearing thresholds (averaged audiogram for both ears), tinnitus loudness (10-point scale), and tinnitus distress scores (TQ). The calculation of the Bayes factors was performed using the 'bayesFactor' Toolbox for MATLAB.

In Study 2, we also obtained decoding accuracies over time for each participant (see Figure 5). For statistical analyses, we used cluster-based permutation *t*-tests (*Maris and Oostenveld, 2007*), with 1000 permutations and a value of p < 0.05 to threshold the clusters in order to account for multiple comparisons. For both tinnitus and control groups, we targeted the pre- and post-stimulus intervals separately. First, we analyzed group comparisons of whether regularity-dependent pre-activations of carrier-frequency-specific information differ between individuals with and without tinnitus. For this, we considered the pre-stimulus interval to perform cluster-based permutation *t*-tests. In a time-generalized manner, we trained the classifier on the random sound sequences and tested on the ordered sequences to take into account the predictability in the ordered sound sequences. Using both entropy conditions, we were able to extract potential regularity-dependent pre-activations of carrier-frequency-specific information. As mid-level entropy conditions were not included in the modified paradigm used in Study 2, we calculated the difference between the ordered and random sequence. Next, we computed group averages and extracted relevant clusters in the pre-stimulus interval as an indicator for regularity-dependent pre-activations. Hence, when comparing the results of Studies 1 and 2, it should be noted that 'neural prediction scores' are based on *β*-coefficients in the former and on differences in the latter case. For our pre-registered statistical analyses (*Reisinger et al., 2024*), we focused on the time window between 470 and 570 ms as training time as this was the time frame of the highest effect in Study 1. We statistically inspected the differences in the clusters between the groups by performing cluster-based permutation *t*-tests and comparing mean decoding accuracies between tinnitus and control groups.

To control for the possibility that group differences attributed to predictive processing by the initial analysis, are due to general differences in processing of tone carrier frequency, we specifically targeted the post-stimulus interval of the random tone sequences. We compared the resulting decoding accuracies over time between groups by implementing cluster-based permutation *t*-tests. Since we expected no difference between groups, we added equivalence testing to strengthen our results (*Walker and Nowacki, 2011*). Furthermore, to ensure that group differences in neural prediction scores are not attributable to variations in decoding bias, we quantified the difference in the average decoding scores of the first and second upper diagonals of the confusion matrix for the random tone sequence in Study 1. This metric reflects the trained classifier's bias toward the neighboring tone frequency in the transition direction of the more ordered sequences. To evaluate group differences, we compared the resulting time-resolved bias scores using cluster-based permutation *t*-tests. To further analyze the effect of tinnitus distress on our findings, we extracted individual values of the short version of the Tinnitus Questionnaire (Mini-TQ) and calculated the mean subjective tinnitus distress for each individual of the tinnitus group. Next, we correlated the individual tinnitus distress

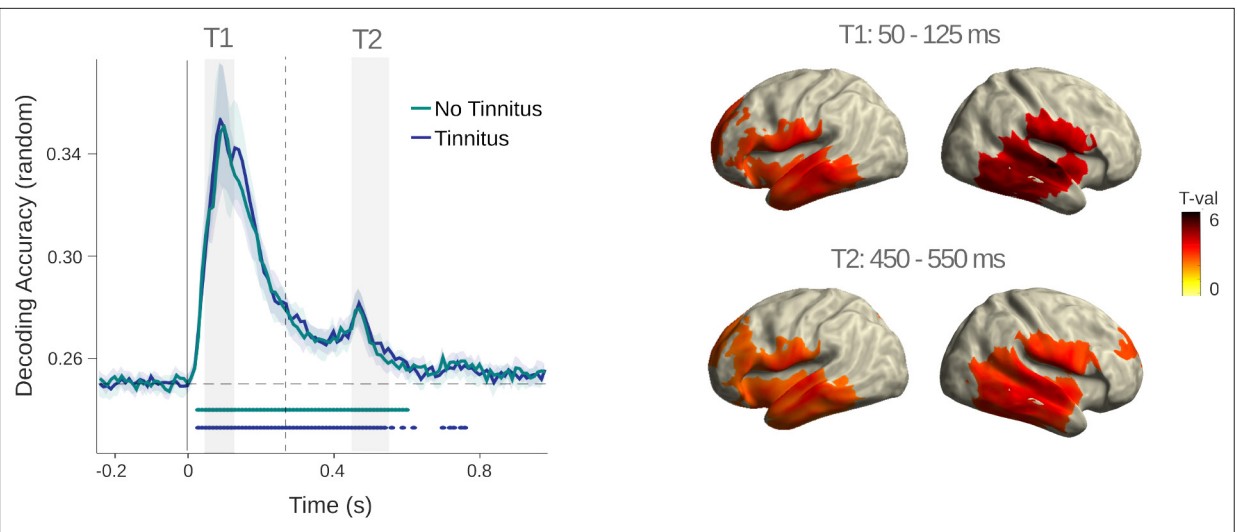

**Figure 2.** Random tone decoding. **Left panel:** Temporal decoding of carrier frequencies in the random sound sequence for the tinnitus and control groups, respectively. In both groups, peak accuracy is reached after ~100 ms following sound onset. Above chance decoding accuracy is observed in a sustained manner up to ~600 ms (p < 0.05, Bonferroni corrected). No differences were observed between the groups. **Right panel:** Source-level depiction of informative activity for different periods: 50–125 ms (T1) and 450–550 ms (T2) after decoded sound presentation. The latter corresponds to the training time interval, yielding pronounced group differences in the condition generalized analysis.

The online version of this article includes the following figure supplement(s) for figure 2:

**Figure supplement 1.** Time-resolved bias score.

values with our previously calculated prediction score (i.e. the difference in decoding between ordered vs random sounds in the pre-stimulus interval). Importantly, information regarding tinnitus distress was not available for all 40 tinnitus subjects but solely 31 subjects were included in this analysis. We therefore excluded the nine subjects that did not complete the Mini-TQ for this specific analysis.

# Results
## Study 1

We will initially report the results of Study 1, followed by the replication of our effects in Study 2. Study 1 was characterized by a relatively large range of stimulus frequencies (200–2000 Hz) and a large number of presented trials. Furthermore, four different levels of entropy within the sound sequences were included, which influence the decoding accuracy were quantified $\beta$-coefficient values drawn from a linear regression analysis.

### General processing of tone carrier frequencies does not differ between tinnitus and controls

Sensor-level MEG data was used to decode the four carrier frequencies presented in the random sound sequence. Since this condition did not contain predictability-related information, it allowed us to compare basic encoding of sound carrier frequencies in individuals with and without tinnitus. Both groups exhibited a rapid increase of decoding accuracy following sound onset, robustly observed at an individual level (*Figure 2*). Above chance (p < 0.05, Bonferroni corrected) decoding accuracy started immediately after stimulus onset in both groups. Interestingly, accuracy transiently increased approximately 100 ms after the subsequent stimulus onset (i.e. 450–500 ms after the to-be-decoded sound). This observation may reflect a sustained activation and reactivation of an auditory short-term memory trace enabling the formation of associations between events in temporal proximity, which is fundamental for subsequent learning of statistical regularities and the formation of an internal model. Importantly, we found no differences between the tinnitus group and the control group. Also, for the same investigation time period, bias scores did not differ between the group (all cluster p's > 0.83, *Figure 2—figure supplement 1*). Therefore, at a basic level, individuals with tinnitus encode and

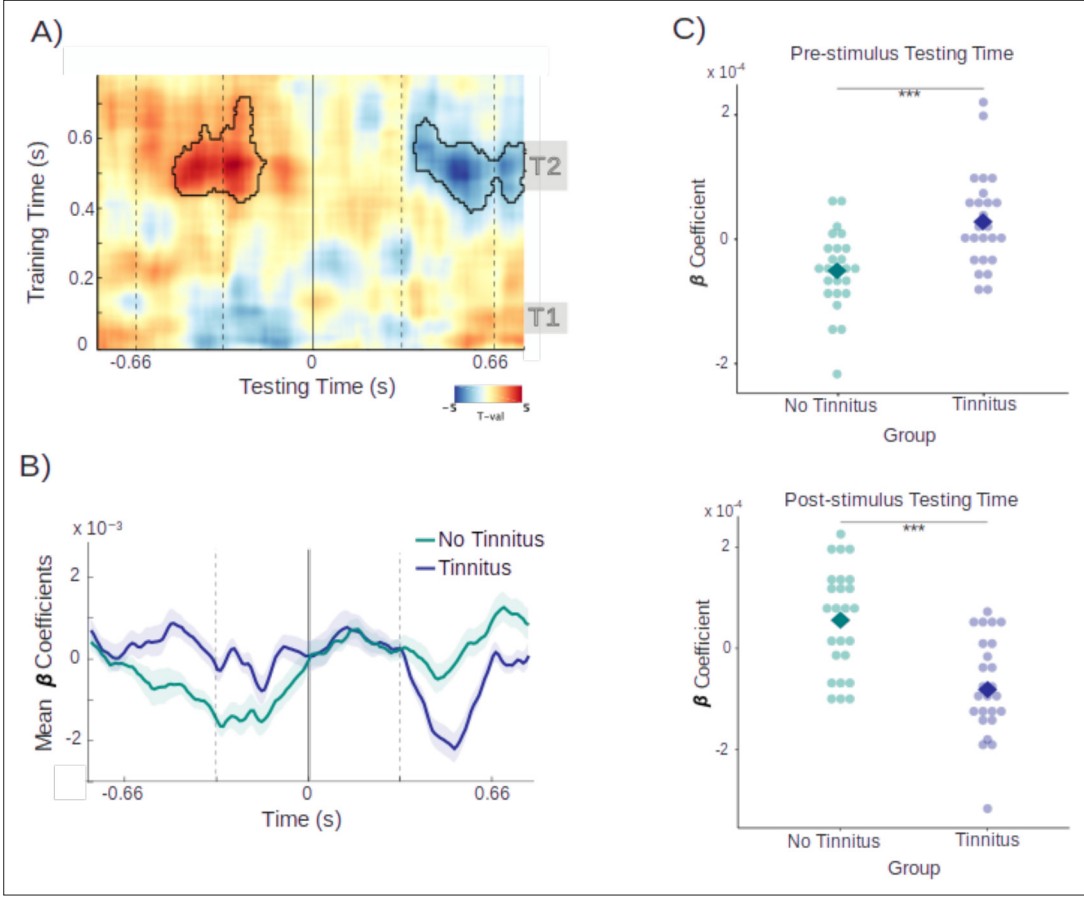

**Figure 3.** Group comparisons before and after sound onset. (**A**) Prediction scores (i.e. group comparison of β-coefficient values) between tinnitus and control groups in the time-generalized matrix. Colors indicate *t*-values and solid black borders delimiting periods of significant difference (p < 0.05, cluster corrected). (**B**) Time courses of β-coefficients averaged over the training time window (450–550 ms). (**C**) Individual β-coefficient values within the pre- and post-stimulus clusters are shown. Asterisks indicate significance levels with p<0.001.

reactivate carrier frequencies equally well to individuals without tinnitus when the context is unpredictable. This means that subsequently reported group differences are due to the manipulation of regularity (i.e. predictability) of the sound sequence.

## Anticipatory frequency information is relatively enhanced in tinnitus

Using the early time window as training time, we identified a positive cluster (p = 0.038) prior to the onset of the to-be-decoded event at approximately –530 to –200 ms (*Figure 3A*). A second, negative cluster (p = 0.05) between 360 and 800 ms was observed, caused by inverse tendencies for the tinnitus and control groups (*Figure 3A*): that is, whereas individuals with tinnitus appeared to quickly deactivate carrier-frequency patterns the more regular the sound sequence became, control individuals reactivate patterns of the decoded sound presented at sound onset upon presentation of new events. We further averaged the time courses of β-coefficients over the training time of 450–550 ms after stimulus onset which showed that whereas individuals with tinnitus exhibited relatively stable processing in the pre-sound period, results for control individuals were marked by a weaker processing the more regular the sound sequences were (captured by the negative β-coefficients, *Figure 3B*). For more detailed insights, individual β-coefficients for the relevant pre- and post-stimulus intervals are shown (*Figure 3C*).

## Pre-stimulus effects are not related to hearing thresholds and tinnitus-related features

Following the demonstration of a marked group difference in activating late carrier-frequency-specific neural patterns as a function of sequence regularity, we tested whether the magnitude of this process was related to subjectively rated tinnitus characteristics as well as audiometric features. Across the full ($n$ = 34) tinnitus sample, we performed Spearman correlation between the averaged $\beta$-regression values corresponding in time to statistically significant anticipatory positive and post-stimulus negative clusters and hearing thresholds, tinnitus loudness, and tinnitus distress. Despite the explorative (liberal) testing without multiple comparison corrections, no significant correlation effects were identified for the pre-stimulus positive cluster for hearing thresholds (rho = −0.06, p = 0.75), tinnitus loudness (rho = −0.06, p = 0.73), and tinnitus distress (rho = 0.11, p = 0.53). All Bayes factors were found to be between 1/3 and 1/10, that is, providing moderate to strong evidence for the null hypothesis. The same tendency was found for the post-stimulus negative cluster for hearing thresholds (rho = −0.13, p = 0.45), tinnitus loudness (rho = −0.01, p = 0.95), and tinnitus distress (rho = −0.14, p = 0.43). Again, Bayes factors ranged between 1/3 and 1/10. The lack of relationships between prediction related neural effects with hearing thresholds add further support to the claim that the effects visible in group analysis are strictly regularity-dependent and not driven by low-level auditory processing.

### Interim discussion

So far, recent research reported only scarce evidence for altered predictive processing in tinnitus, which additionally has been rather indirect, that is did not investigate feature specificity of neural activity patterns. Since predictions are usually *about* something, experimental approaches need to take this into account. In a first necessary step, we compared individuals with chronic tinnitus and controls without tinnitus, utilizing a previously established paradigm (*Demarchi et al., 2019*) that allowed us to scrutinize feature specificity of predictive processes in the auditory system at high temporal resolution. Our main findings are: (1) basic processing of carrier frequencies are not altered in tinnitus; (2) with increasing regularity of the sequence, individuals with tinnitus show relatively enhanced predictions of frequency information; (3) the effect is not related to hearing thresholds and tinnitus distress or loudness in this sample.

A major limitation of this Study 1 was the insufficient control for hearing loss as a potential confound of the results. Audiometric thresholds were not assessed consistently for all control participants and hence hearing status was not matched between individuals with and without tinnitus. This motivated the implementation of a second study, aiming for an overall, rigorous matching of age, sex, and – most importantly – hearing loss. Data collection and analyses of this Study 2 were embedded in the process of a registered report (*Reisinger et al., 2024*). To allow a more comprehensive interpretation of our effects against the background of two independent samples with highly similar study designs, we will report our second findings in the next section, followed by an overall discussion.

## Study 2

As mentioned before, Study 2 was by itself already recommended as a registered report by the Peer Community In Registered Reports (https://osf.io/8bv29, *Chambers, 2024*). In a first step, we submitted our proposed sample size, hypotheses and the exact analysis protocol. This stage 1 report was peer-reviewed and accepted prior to accessing the data. Afterwards, analyses were conducted in line with the pre-registered methods and interpreted with regard to our hypotheses. The finalized stage 2 manuscript was peer-reviewed a second time and recommended as we did not deviate from our proposed stage 1 report.

Study 2 consisted of 40 individuals with tinnitus as well as 40 controls without tinnitus who were matched in terms of age, sex, and hearing loss. Unlike Study 1, the experimental design for this second study varied in terms of exact stimulus set and duration (see *Figure 1*). Sound carrier frequencies ranged between approximately 440 and 1043 Hz which was significantly narrower compared to the first design (200–2000 Hz). Furthermore, sound sequences were either random or ordered, without two additional variations in regularity (i.e. midminus and midplus). The duration was as well reduced by including less trials per condition. We will discuss the influences of these deviations between study designs below.

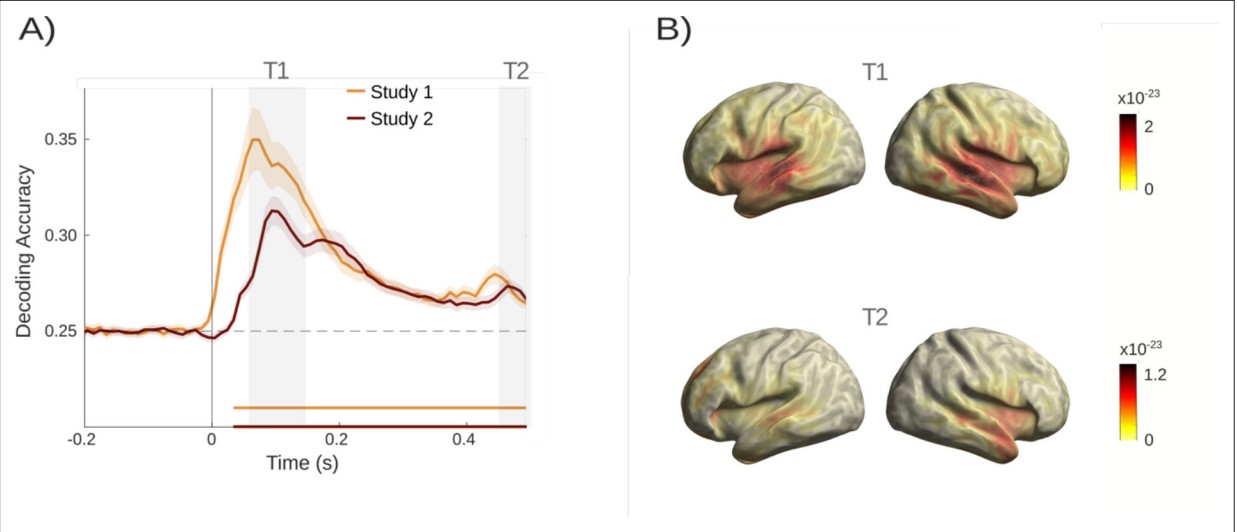

**Figure 4.** Random tone decoding accuracies. (**A**) Comparison of random tone decoding accuracies between the two studies. Gray areas indicate relevant peaks in decoding accuracies. (**B**) Source plots demonstrating activity in the auditory cortex for both time windows in Study 2.

## Decoding of tone carrier frequencies from MEG data

As for Study 1, our analysis critically relied on the ability of our classifier to reliably decode carrier-frequency information from MEG data. As illustrated in *Figure 4A* (red line), this is clearly the case for the entire sample ($t_{260}$ = 8.40, 95% CI [0.0092, 0.0148], p < 0.001), with a rapid increase of decoding performance peaking at ~100 ms, and a slow decrease however with above chance decoding performance lasting throughout the entire post-stimulus period, including a small increase following the subsequent tone presented at 333 ms. Consecutive source analyses demonstrated strong activations in the temporal areas in both hemispheres for the first time window which were still present in the later time window – however less pronounced and more lateralized (*Figure 4B*).

*Figure 4A* further includes decoding patterns for both studies. Descriptively, the red line depicting values of our Study 2 resembles the pattern of Study 1 very much, as shown as the orange line in *Figure 4A*. There, we also observed similar peaks in decoding accuracies around 100 ms (T1) and 450 ms (T2) after target stimulus presentation. However, when comparing the outcomes of both studies, some differences can be noticed. First, a ~30 ms delay can be seen in Study 2 and more importantly,

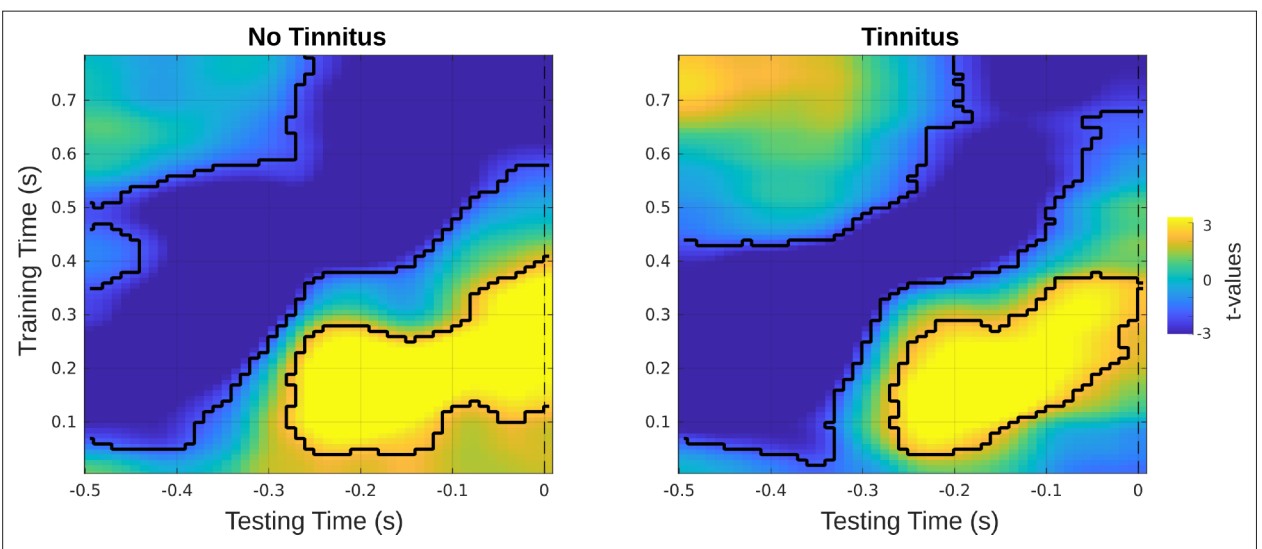

**Figure 5.** Time generalization matrices over training and testing time for the tinnitus and control groups. Black masks indicate significant difference clusters between the ordered and random sound condition.

accuracies were significantly lower ($t_{260}$ = 2.16, 95% CI [0.0005, 0.0109], p = 0.031). These differences are most likely due modifications of the paradigm, and particularly the much narrower sound frequency range in Study 2 (1.3 octaves) as compared to Study 1 (3.3 octaves), making the decoding more challenging. This is of relevance when comparing and interpreting differences of group effects, presented, and discussed further below.

### Anticipatory sound carrier-frequency information is relatively increased in tinnitus

As in Study 1, we again trained the classifier on the random tone sequence, and applied it to both regularity levels, to assess whether this changes decoding accuracies over time. *Figure 5* shows the constructed time generalization matrices for the tinnitus and non-tinnitus group. When training on the post-stimulus interval and testing on the pre-stimulus interval, the matrices descriptively indicated differences between the groups in the processing of ordered versus random sounds. In the training

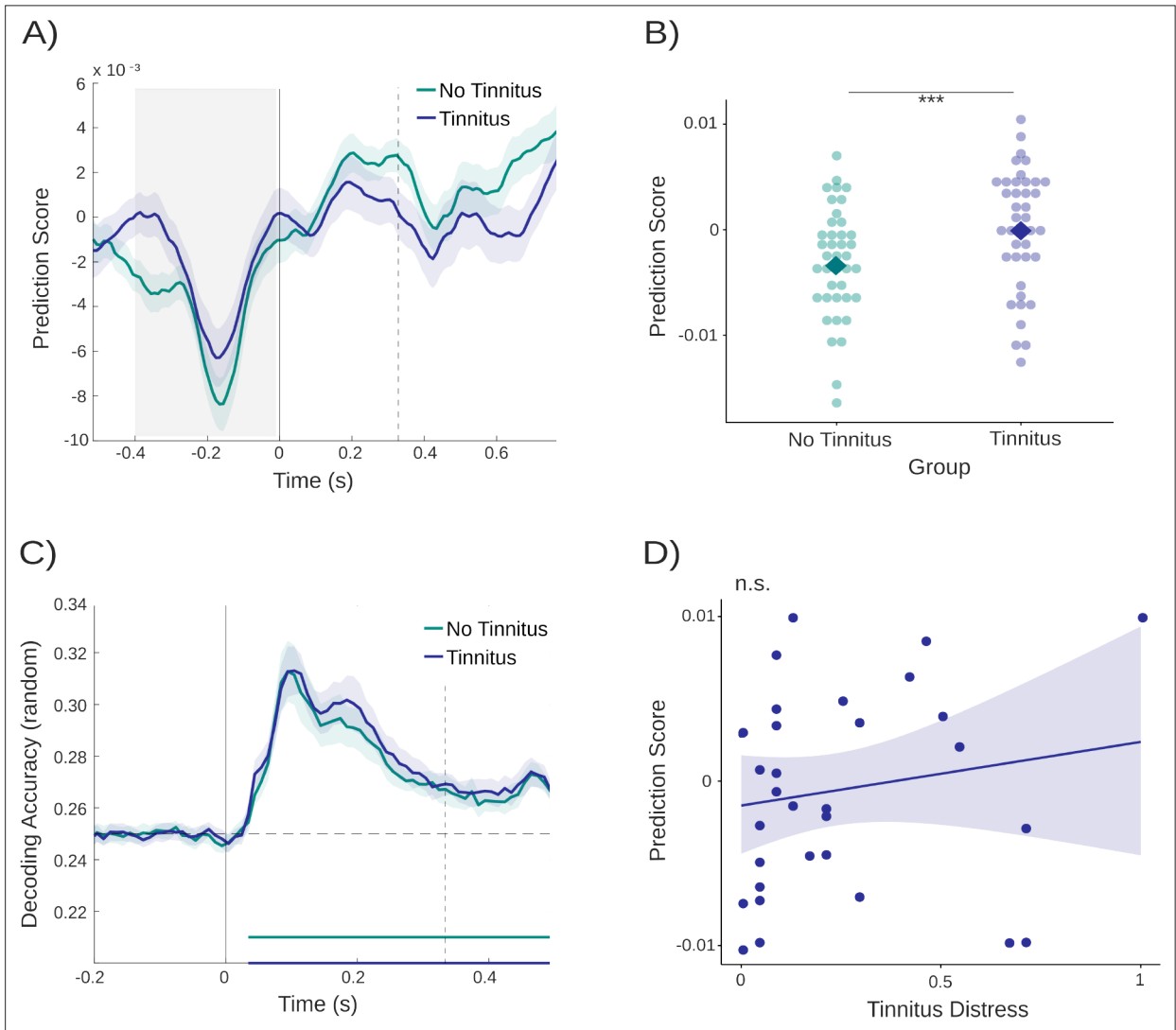

**Figure 6.** Differences in decoding accuracies between groups and correlations with tinnitus distress. (**A**) Prediction scores with the second time window (470–570 ms, i.e. the relevant time window in Study 1) as training time. The gray area depicts the time window used for statistical analyses and vertical lines indicate sound onsets. (**B**) For illustration purposes, depiction of individual prediction scores at the time point of the most pronounced group difference (–380 ms). Squares indicate the mean of each group. Asterisks indicate significance levels with p<0.001. (**C**) Random tone decoding accuracy over time for both groups. Vertical lines indicate sound onsets, and the dashed horizontal line depicts decoding at chance level. (**D**) Positive, non-significant correlation between prediction scores and tinnitus distress reported in the tinnitus group.

interval around 500 ms and the testing interval around –400 ms differences in decoding ordered versus random sounds appeared to be more pronounced in the group without tinnitus.

To focus our analysis on the replication we targeted a 470- to 570-ms time window for training, which yielded the strongest effect in Study 1. When testing on the pre-stimulus interval between –400 and 0 ms, one-sided cluster-based permutation $t$-tests revealed a significant positive cluster ($t_{sum}$ = 26.2, p = 0.046). In line with our prediction, this indicated relatively higher decoding accuracies in the tinnitus group and therefore replicated the central finding in Study 1 (*Figure 6A*). The effect appeared to be most pronounced in the time window between –410 and –310 ms. In this time frame, relevant $t$-values ranged from $t$ = 1.90 to $t$ = 2.75 with significant p-values between p = 0.005 and p = 0.036, indicating stronger differences in decoding accuracy between ordered and random sound sequences (i.e. higher prediction scores) in the tinnitus group.

The statistical difference, however, is not informative about what is driving the effect. *Figure 6A* is suggestive that group differences could be mainly driven by below-chance decoding in the control group, as already observed for Study 1 (see *Figure 5*). To follow this up, we extracted the individual values for the tinnitus and control groups at the time point with the greatest group difference (around –380 ms) and displayed them to further illustrate the difference between the two groups on an individual level (*Figure 6B*). Comparing the prediction scores against zero within each group indicated non-significance for the tinnitus group with $t_{78}$ = 0.063, 95% CI [–0.0017, 0.0018], p = 0.950. In the group without tinnitus, however, prediction scores significantly varied from zero ($t_{78}$ = –3.65, 95% CI [–0.0047, –0.0014], p < .001). This indeed indicates that prediction score differences between tinnitus and control groups are driven by below-chance accuracy in the control group at these late training time intervals. Interestingly, effects showed a similar pattern in both studies.

## General processing of tone carrier frequencies does not differ between tinnitus and controls

As in Study 1, it is important to ensure that reported group differences cannot be 'trivially' explained but are genuinely due to differences in predictive processing. By design, the confounding impact of hearing loss was eliminated. Nevertheless, it is possible that one group exhibits superior processing capacities of carrier-frequency information. Results showed that, averaged over time, no significant difference between the participants with and without tinnitus was observed using an independent sample $t$-tests ($t_{260}$ = 0.795, 95% CI [–0.0024, 0.0056], p = 0.428). As absence of evidence does not equal evidence of absence of an effect, we included an additional equivalence test (*Walker and Nowacki, 2011*). This yielded a significant result of $t_{73.93}$ = –19.08, p < 0.001, indicating equivalence between the two groups (*Figure 6C*). Therefore, the group difference reported above can indeed be attributed to an altered (anticipatory) processing of tones presented at different levels of regularity.

## Pre-stimulus differences in ordered and random tone sequences are not related to tinnitus distress

Observing a group difference (tinnitus vs no tinnitus), as reported previously, could also be driven by non-specific effects, such as the perceived distress of individuals rather than due to the condition directly. Thus, building on our previous study, Spearman's rank correlation revealed no significant relation between tinnitus distress and decoding accuracy with $r$(30) = 0.25, p = 0.168 (*Figure 6D*). Together, with the result of Study 1, our results strongly support the notion that unspecific distress due to tinnitus is not a good explanation for the identified differences in decoding accuracy. We additionally compared tinnitus distress values assessed by the mini-TQ (*Goebel and Hiller, 1992*) between Studies 1 and 2 to detect potential differences between the samples, however, results of the Welch's $t$-test were not significant with $t$(30.7) = 1.27, p = 0.214.

### Exploratory results

Using two independent studies, we were able to successfully replicate our findings indicating altered (anticipatory) processing of tones in tinnitus. However, group differences appeared to be less pronounced in Study 2. As mentioned previously, the study designs varied to some degree between the two studies and, therefore, we aimed to additionally identify relevant aspects that potentially affected the strength of our findings. Specifically, we further investigated the effects of stimulus frequencies and trial numbers, as well as the influence of hearing loss.

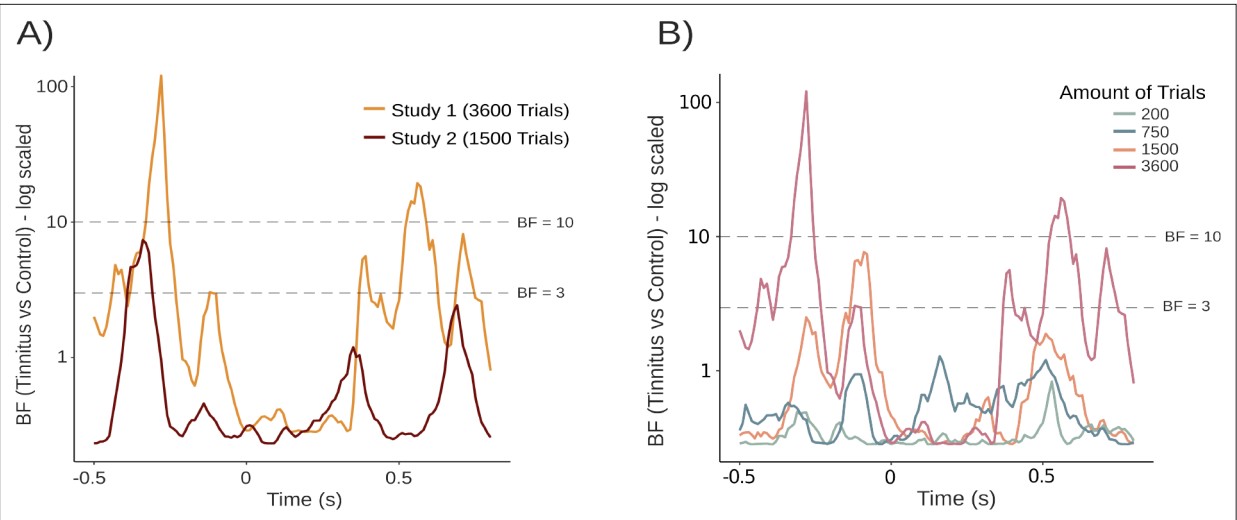

**Figure 7.** Exploratory analyses. (**A**) Bayes factor analyses of the differences in decoding accuracies between the tinnitus and control group for both datasets indicated higher effects in Study 1. (**B**) Different numbers of trials used for decoding were drawn from Study 1. Comparing these, Bayes factor analyses demonstrated higher and relevant effects solely in datasets with a high amount of trials. Horizontal, dashed lines in both figures indicate Bayes factor supporting a difference between samples (see *Schönbrodt and Wagenmakers, 2018*).

## Strength of group effects differs between the two studies

As demonstrated earlier, decoding accuracy for carrier frequencies in the random tone sequences was overall stronger in Study 1 as compared to Study 2. Also, the core group effect appears to be smaller compared in Study 2 (see *Figure 6A*). To put this descriptive impression into quantitative terms, in a first step, we compared the time-varying overall difference in decoding accuracy between ordered and random tones for the tinnitus and control group in both studies. Bayes factors demonstrated relevant effects in both datasets in the pre-stimulus interval at around –400 ms, however, values are much higher in Study 1 as compared to Study 2 (maximum BF 120.7 vs 7.4). In the post-stimulus interval, the second dataset does not reach a Bayes factor over at least 3, indicating no difference between the tinnitus and control group. For the first dataset, higher values were found as well in the post-stimulus interval, starting at around 400 ms (*Figure 7A*).

It is worth noting again that the samples for these two studies are completely independent and in Study 2 rigorous control of hearing loss was undertaken. While this may have a diminishing impact on the effects linked to predictive processing, other factors related to paradigm adaptations may have played a more decisive role. To guide future design decisions, it is therefore important to understand the influence of relevant experimental choices on the group difference effect. We have previously noted that the setting of the tone frequency boundaries was much broader in Study 1 (1.3 vs 3.3 octaves). This was initially done to ensure that the highest frequency in participants falls into a normal hearing range. The closer frequencies come along with more similar neural patterns that are more difficult to classify. This becomes apparent when comparing the decoding of pure tones, with significantly higher accuracy in Study 1 as well as earlier peaks. As the classifier is trained on the random sequence and applied to the ordered sequence, a less accurate classifier in Study 2 likely decreased the sensitivity of correctly classifying patterns in the pre-stimulus period. Overall, the choice of frequencies in Study 2 likely leads to an underestimation of the true group differences.

## Group differences depend on the number of trials available for training the classifier

Apart from the choice of carrier frequencies, another obvious difference between the two studies was the difference in trial numbers. The very high trial number in Study 1 was due to maintaining the exact study design of *Demarchi et al., 2019* that had one focus on investigating omission periods. As omission responses were not of interest in the second design, and the experiment was embedded in further testing, we decided to implement less trials in Study 2. While reducing measurement time significantly, our design choice also meant that fewer trials could be used to run the decoding

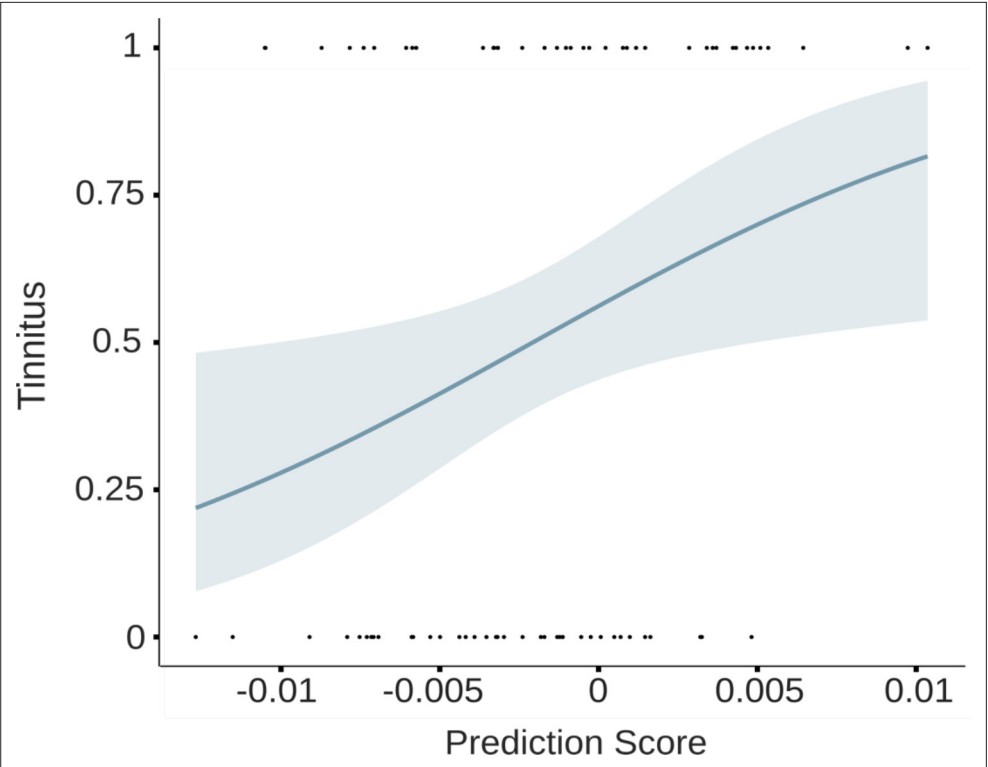

**Figure 8.** Logistic regression indicates that higher prediction scores significantly predict the status as tinnitus participant (*P*(Tinnitus) = 1).

analyses. To get an indication of how the quantity of trials impacts the strength of observed group differences, we used the data of Study 1 which had 3600 trials for each condition. We randomly drew various numbers of trials, ranging from 200 to 3600 trials, for each carrier frequency and recalculated the group difference effects expressed in Bayes factors (see *Figure 7B*). Bayes factors greater than 3 are considered supportive of a difference (*Schönbrodt and Wagenmakers, 2018*). These values were solely found for subsets with higher numbers of trials (i.e. 3600 or 1500 trials). A lower number of trials like 750 or 200 trials showed no relevant group differences. Therefore, the number of trials highly impacts the strength of effects and was likely responsible for the smaller effects in Study 2 since solely 1500 trials per condition were included.

## Group differences in predictive processing are not confounded by hearing loss

As described previously, a major challenge for the interpretation of group differences in Study 1 was the lack of control for hearing loss, which most likely differs between the two groups when not explicitly taken care of. In Study 2, by design, we diligently matched both groups for overall level of hearing loss, hence making it unlikely that this factor accounts for the group difference reported in both studies. To provide further corroborative evidence, we decided to analyze the group difference effect using a complementary approach. For this purpose, we used logistic regression to analyze the presence of tinnitus as a dependent variable, and prediction scores in the pre-stimulus interval as independent variable, controlling for hearing loss by including mean hearing ability as a covariate. In a first model, predicting tinnitus solely by prediction scores, we could identify this as a significant predictor in Study 2 ($b$ = 27.96, SE = 11.53, p = 0.018). As expected, when adding mean hearing loss to the model, it does not significantly predict the presence of tinnitus ($b$ = 0.007, SE = 0.004, p = 0.076). Importantly (see *Figure 8*), controlling for the influence of hearing loss, the prediction scores still significantly predicted whether participants experience tinnitus ($b$ = 24.63, SE = 11.51, p = 0.036). We included an additional odds ratio analysis, resulting in OR = 1.31, indicating an increase of 1 sd in the prediction score increases the odds of having tinnitus by 31%. Reversing this model,

we also analyzed whether the presence of tinnitus can predict prediction scores. This model was significant as well ($b$ = 0.003, SE = 0.001, $p$ = 0.018), indicating a statistically relevant influence of tinnitus on the prediction scores and further supporting our previous findings. As these logistic regression models were computed using an average hearing score computed over the frequencies 500, 1000, 2000, and 4000 Hz (i.e. PTA-4, see e.g. *Lin et al., 2011*; *Ozdek et al., 2010*), we questioned whether hearing loss in higher frequencies influenced our effects. We therefore computed an additional logistic regression including also the PTA values of 6000 and 8000 Hz. In this analysis, hearing loss was not a significant predictor of tinnitus but rather showed a trend with $b$ = 0.211, SE = 0.111, $p$ = 0.062. Prediction scores, however, remained a significant predictor of tinnitus even after including high-frequency hearing loss ($b$ = 0.232, SE = 0.111, $p$ = 0.040). In this analysis, odds ratios indicated an increase of 26% in the odds of having tinnitus with a 1 sd increase in the prediction score. Overall, this analysis strongly supports the notion that the main effect genuinely reflects a process related to the experience or statistical risk of experiencing tinnitus.

## Discussion

Despite strong efforts, neuroscientific tinnitus research in humans is in dire need of robust findings that would enable a deeper understanding of the mechanisms of this condition (*Reisinger et al., 2023*). Large numbers of studies, especially using EEG or MEG, have focused on spectral characteristics of resting-state activity. Apart from lacking empirical robustness of various 'neural correlates', most of the underlying conceptual assumptions, often based on hyperactive or hypersynchronous neural ensembles, are insufficient in explaining distinct phenomena (*Sedley, 2019a*). A conceptual challenge is observing why some individuals with hearing loss develop tinnitus, whereas others with comparable hearing loss do not. To overcome these issues, the application of predictive processing models to tinnitus has become particularly popular (*Sedley et al., 2016b*). Distinct predictive processing patterns could for example, either develop within an individual in contributing to chronification of tinnitus (e.g. shift of 'default prediction' from silence to sound; *Sedley, 2019a*). Alternatively, they could be conceived as sensory processing style, making certain individuals more vulnerable to develop tinnitus under certain conditions (e.g. hearing loss, aging), a notion reminiscent of the 'strong prior' hypothesis of hallucinations (*Corlett et al., 2019*). Hence, the direction of the effect remains unclear and alternative explanations, such as a third variable being responsible for predictions and tinnitus development, cannot be excluded with certainty. Furthermore, even if altered predictive tendencies were to be found, there could be various possibilities of exactly how they could be altered to contribute to the onset or persistence of tinnitus. In any case, any more conclusive claims would require longitudinal data, ideally with a tinnitus-free baseline. As such research is challenging to implement, especially in humans, we first focused in this work on finding cross-sectional group differences between individuals with and without tinnitus.

### The relevant group differences were successfully replicated and cannot be explained by general tone processing, hearing loss, or tinnitus distress

In Study 1, using a simple MEG paradigm (*Demarchi et al., 2019*), we were able to show relatively enhanced tone frequency-specific pre-activation in tinnitus. Study 2 included in this work overcame a key limitation of Study 1 by rigorously controlling for hearing loss. This was unfortunately not assessed in Study 1, leaving the possibility open that predictive processing effects were due to hearing loss rather than tinnitus. By showing that the key effect remains robust in another study as well and despite some important changes in the paradigm, it provides robust evidence for a role of altered predictive processing in understanding tinnitus. We refer to this altered pre-stimulus activity pattern as a prediction score in tinnitus, indicating deviations between random and predictable sounds that are specifically found in tinnitus patients. By now, this prediction score specifies merely a broader concept than a concrete numerical score. Influential factors and underlying mechanisms are still not fully understood, and it is therefore not applicable yet to determine a concrete value to quantify neural predictions.

Using a late training time interval (~470–570 ms), the prediction score was significantly higher in the tinnitus group in the pre-stimulus time window of interest and especially pronounced ~400 ms prior to stimulus onset. Source analyses revealed auditory activations in both hemispheres for these

two time frames, in a comparable manner in both datasets. Importantly, we showed in both studies that decoding accuracy as well as bias scores did not differ between groups. This means that any group difference in terms of prediction scores needs to be attributed to altered processing of the regularity of the sound sequence. In a complementary analysis, we used our prediction score in addition to hearing loss magnitudes as predictors of tinnitus in a logistic regression. Prediction related pre-activation levels were informative whether participants perceived tinnitus, also when statistically controlling for hearing loss. However, it has to be mentioned that we calculated hearing loss based on the PTA results of the frequencies between 500 and 4000 Hz. This does not reflect hearing impairments like high-frequency hearing loss or hidden hearing loss (i.e. hearing difficulties despite a normal audiogram, *Liberman, 2015*). As for hidden hearing loss, we were not able to draw conclusions regarding our effects since this concept of hearing damage is difficult to measure objectively, especially in humans. However, we included an additional logistic regression expanding the frequency range up to 8000 Hz and again, hearing loss did not substantially impact the prediction score as an informative tinnitus predictor.

However, as groups can differ with respect to many aspects, this does not mean that the effect reported in both studies is specific to the perception of the phantom sound. One possibility is that tinnitus goes along with some distress, which could be the actual cause for the pre-stimulus effect. In this case, we would expect a correlation of this effect with tinnitus related distress. However, the correlation was not significant in both studies. Thus, distress is not a potent candidate in explaining the reported effect.

In conclusion, the results provide strong support for the notion that aspects of predictive processing play some – to be defined – role in understanding tinnitus.

## The group differences in the prediction score are driven by above-chance deactivation in non-tinnitus individuals – implications for understanding tinnitus

Overall, relatively enhanced prediction scores in the tinnitus group were shown in both studies, though we still do not know the underlying mechanisms of this effect. These in relative terms stronger pre-activations could indicate a higher predisposition or vulnerability to develop tinnitus, for example in terms of the strong prior hypothesis by *Corlett et al., 2019*. Within this framework, it is assumed that hallucinations or phantom perceptions arise when prior beliefs disproportionately influence perceptual inferences, leading to perceptions without sensory input (*Friston and Kiebel, 2009*; *Powers et al., 2017*). Therefore, vulnerability to develop tinnitus might be due to stronger individual predictions or pre-activations of future sounds, that is higher prediction scores. However, it is clear that the group differences seen in both studies with respect to pre-activation patterns stems from below chance decoding for the control group, that is, a significant deactivation of anticipated carrier-frequency-specific neural patterns at late training time intervals. Note that for the early training time intervals (<300 ms) patterns between both groups are highly similar, generally marked by enhanced pre-activation of anticipated tone frequencies (see e.g. *Figure 5*). This does not fit to a very simplistic notion of generally enhanced predictive processing in individuals with tinnitus.

For both studies the pre-stimulus group effect was identified at late training time intervals. These refer to patterns, trained on the random sound sequences, which are visible as small late peaks (see *Figures 2 and 6C*). As they follow the onset of the subsequent tone, in the random tone sequence they are indicative of a reactivation of carrier-frequency-specific information and potentially neural processes involved in forming associations between events and hence formation of auditory memories. Applied to ordered sequences, below chance decoding indicates a meaningfully lacking activation of the target carrier frequency in non-tinnitus individuals for these putatively higher-order auditory processes.

Even though the group differences appear most pronounced prior to target stimulus onset, it is more difficult here to speak of pre-activation effects, as in an ordered sequence the previous and subsequent tones are more likely to be *a specific other* carrier frequency (as determined by the transition matrix). Indeed, while below chance decoding drives the pre-stimulus effect in both studies, the opposite effect is present following the subsequent tone(s) in Study 1 (see *Figure 3B*) and – albeit not significant – a descriptively similar patterns can be seen in Study 2 (see *Figures 6A and 7A*). The

absence of a clearer effect in the post-stimulus period in Study 2 could be attributable to the design limitations presented above and discussed below.

Since pre-activation of early training time windows are similar in both groups, we speculate that once the regularity of the sound sequence is learned, individuals with tinnitus downregulate these higher-level auditory processes. Translated into more conceptual terms, this could mean that individuals without tinnitus use associative contextual information to continuously 'monitor' internal models and these associative processes are less engaged in tinnitus once an internal model has formed. Another possible explanation compatible with our effects is that, in line with predictive processes, predictions of neural representations of stimuli are pre-deactivated to minimize responses to expected stimuli. In tinnitus, this sensitivity to predictable stimuli could therefore be enhanced. People with a tinnitus predisposition are less likely to suppress that signal on account of its predictabilities, and therefore that signal is more likely to become perceptible initially. Downregulations of higher-level auditory processes for predicted stimuli can then indicate that auditory predictions become more strongly represented, and therefore harder to change through competing sensory evidence. These fundamental differences in auditory processing 'styles' could also be an interesting conceptual bridge to addressing why tinnitus becomes chronic and so therapy resistant in some individuals.

As indicated above, these interpretations remain speculative at this stage and requires further investigation. We acknowledge that there are other possibilities on how to interpret the below-chance decoding and therefore encourage future work to implement novel approaches to gain more insights into the exact underlying mechanisms driving this effect. An increased focus on hippocampal regions, for example, in fMRI, patient, or animal studies, could be a worthwhile complement to our MEG work, given the outstanding relevance of medial temporal areas in the formation of associations in statistical learning paradigms (see e.g. *Covington et al., 2018*; *Paquette et al., 2017*; *Schapiro et al., 2016*).

## Implications for future studies using the tone regularity paradigm

Despite the successful replication of our core findings in Study 2, it is noticeable that group differences effects were weaker compared to Study 1. This becomes particularly evident when comparing time-varying Bayes factors for the effects from both studies (*Figure 7A*). While the strongest Bayes factors in the pre-stimulus period reached values above 100 in Study 1, they reached a value of around 7 in Study 2. Understanding which aspects contribute to this decline in effect size is relevant for future works.

An obvious assumption could be that the control for hearing loss led to a decrease in the magnitude of the effect. Even though this possibility cannot be completely excluded, we deem it very unlikely as hearing loss was neither associated with the prediction score in Study 1 (audiometry was available for the tinnitus group) nor in Study 2 assessed in the logistic regression analysis.

Aside from controlling for hearing loss, Study 2 deviated in some respects from Study 1, which are more likely to influence the strength of the effect. First, we chose a narrower frequency range at lower frequencies (1.3 vs 3.3 octaves) for the second study design, with the intention to assure that carrier frequencies fall into the normal hearing range. However, closer frequencies lead to more similar neural patterns hence making it more challenging for a classifier to decode tone frequencies. The lower decoding accuracy for carrier frequencies and slightly shifted decoding peaks in the random sequence in Study 2 are a testimony to this issue. As these classifiers were also applied to the pre-stimulus periods of the ordered sequence, less reliable decoding likely obscured group differences to some extent.

Second, a difference between the two datasets is the number of trials included in the decoding analyses. In Study 2, less than half the trial number was used to train the classifier as compared to Study 1. The generous number of trials in Study 1 allowed us to quantify the influence of trial number on the group difference for the core predictive processing effect. By repeating the analysis pipeline for a varying amount of randomly sampled trials we observed that Bayes factors decreased with fewer trials and were pointing to no effects once going below 1000 trials per carrier frequency. Study 2 included 1500 trials per carrier frequency, which is likely on the boundary of finding predefined group effects.

Third, both studies used individual sound intensity levels to ensure a comfortable listening situation for the participants. These differences in sound intensity levels are, however, a potential confound in the experimental design as well since sound intensity can have an impact on neural responses

(*Thaerig et al., 2008*). Although in this design, we expect the intensity levels balanced equally to the hearing loss of the participants (which did not differ between groups), and basic decoding of sound frequency did not differ in both studies, we are not able to ultimately exclude the sound intensity level as a driver of our effects. Future studies should include a perceived loudness matching for each frequency and should compare the adapted sound intensity values between each group or integrate them into the analysis (e.g. using the logistic regression approach in *Figure 8*).

Overall, our work clearly underlines the true presence of differences, in terms of predictive processing, between individuals with and without tinnitus. At the same time, distinct design choices impact the strength of the effects which is not only apparent in the present work but was also reported recently by *Yukhnovich et al., 2024*. Further to controlling for basic variables (age, sex, and hearing loss), future studies using our paradigm and analysis approach should opt for a broad frequency spacing (>2 octaves) and ideally more than 2000 trials per carrier frequency in the random sequence. These recommendations are likely even more important for efforts of testing this paradigm using EEG, which normally comes with inferior data quality as compared to MEG.

Furthermore, apart from the methodological specifications mentioned above, attention appears to be relevant in tinnitus as well, both in the generation and the formation of predictions (*Durai et al., 2018*; *Roberts et al., 2013*; *Sedley et al., 2016b*). In the current work, we used passive listening tasks including a movie to reduce attentional focus on the presented stimuli. Therefore, we cannot draw conclusions whether differences in attention had an influence on the effects. Future studies should include more manipulations of attention to investigate its relevance. Additionally, we rigorously controlled for hearing loss in Study 2, however, pure-tone audiometric testing was solely performed up to 8 kHz and we were therefore not able to draw conclusions regarding hearing impairments in higher frequencies and their influence on the effects. Moreover, we did not screen our participants for hyperacusis. This hypersensitivity to mild sounds is widely correlated with the sensation of tinnitus and underlying neural mechanisms are potentially intertwined with tinnitus processes (*Schilling et al., 2023*; *Yukhnovich et al., 2023*). Screening for hyperacusis in future work can therefore reveal more details on participant characteristics influencing predictive processing.

In both studies, tinnitus distress was not correlated with the reported prediction effects. Nevertheless, tinnitus can also be characterized by other features such as its loudness, pitch or duration which were not included in the experimental assessment. Additionally, we solely used a short version of the Mini-TQ (*Goebel and Hiller, 1992*) in Study 2, which did not allow us to relate prediction scores to subscales like sleep disturbances which potentially influence cognitive functioning and thus predictive processing. Next to sleeping disorders and distress, tinnitus is often also accompanied by psychological comorbidities such as depression or anxiety (*Langguth, 2011*) which are potential confounds of the results. For the work described in this manuscript the replicability of the core finding was of main importance. More studies are needed taking into account to assess relate the prediction patterns in more detail to aspects of tinnitus sensation and distress.

## Conclusions

The present work poses the rare instance of a replicable effect in the human neuroscientific tinnitus literature, particularly of the core effect of an enhanced prediction score in tinnitus. The results of Study 2 bear particular relevance by excluding hearing loss as a confounding explanation for the observed differences in anticipatory neural information in tinnitus. Bolstered by the recommendation as a registered report, the robustness of the effect despite changes of some design details is worth emphasizing. For example, the narrower frequency range in the pre-registered second study showed that the effects are to some extent generalizable and not bound to for example, higher-frequency falling into a tinnitus-frequency range. However, some of the design details likely had an adverse impact on the effect size. Concrete suggestions for future studies using our paradigm have been offered.

We concluded that the core pre-stimulus predictive processing effect is caused by a below-chance decoding accuracy in the non-tinnitus group. This indicates that our original notion of a more generally enhanced prediction score in tinnitus individuals is likely too simplistic. We have offered some speculations, however, further studies using complementary methods and designs will be needed to better understand this effect.

## Acknowledgements

LR was funded by the Land Salzburg ("Hidden Hearing Loss", 20204-WISS/225/288/4-2021).

## Additional information

### Competing interests

Jonas Obleser: Reviewing editor, *eLife*. The other authors declare that no competing interests exist.

### Funding

| Funder | Grant reference number | Author |
|---|---|---|
| Land Salzburg | 20204-WISS/225/288/4-2021 | Lisa Reisinger |

The funders had no role in study design, data collection and interpretation, or the decision to submit the work for publication.

### Author contributions

Lisa Reisinger, Data curation, Formal analysis, Visualization, Writing – original draft, Writing – review and editing; Gianpaolo Demarchi, Conceptualization, Formal analysis, Supervision, Validation, Visualization, Writing – original draft, Writing – review and editing; Jonas Obleser, William Sedley, Juliane Schubert, Quirin Gehmacher, Winfried Schlee, Validation, Writing – review and editing; Marta Partyka, Data curation, Formal analysis, Investigation, Writing – original draft; Sebastian Roesch, Nina Suess, Eugen Trinka, Writing – review and editing; Nathan Weisz, Conceptualization, Supervision, Funding acquisition, Validation, Writing – original draft, Writing – review and editing

### Author ORCIDs

Lisa Reisinger ● https://orcid.org/0000-0002-8125-850X
Jonas Obleser ● https://orcid.org/0000-0002-7619-0459
Juliane Schubert ● https://orcid.org/0000-0002-2536-6522
Nathan Weisz ● https://orcid.org/0000-0001-7816-0037

### Ethics

Written informed consent was obtained by every subject. The experimental protocols were approved by the ethics committee of the University of Salzburg (EK-GZ: 22/2016 with Addenda).

Reviewer #1 (Public Review): https://doi.org/10.7554/eLife.99757.4.sa1
Reviewer #2 (Public review): https://doi.org/10.7554/eLife.99757.4.sa2
Author response https://doi.org/10.7554/eLife.99757.4.sa3

## Additional files

### Supplementary files

MDAR checklist

### Data availability

Data is available at https://gin.g-node.org/lisareisinger/tinnitus_predictions.

The following dataset was generated:

| Author(s) | Year | Dataset title | Dataset URL | Database and Identifier |
|---|---|---|---|---|
| Reisinger L | 2023 | Tinnitus Predictions | https://gin.g-node.org/lisareisinger/tinnitus_predictions | G-Node GIN, tinnitus_predictions |

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
