## [Editor Report · eLife Assessment]

This **important** work presents two studies on predictive processing in subjects with and without tinnitus, matched for age, sex and hearing loss. These studies together provide **compelling** evidence for an enhanced predictability of upcoming sounds in regular sequences in EEG data recorded from tinnitus subjects. This work will be of interest to researchers, especially neuroscientists, in the tinnitus field and beyond.

---

## [Referee Report · Reviewer #1 (Public Review)]

This work presents a replicable difference in predictive processing between subjects with and without tinnitus. In two independent MEG studies and using a passive listening paradigm, the authors identify an enhanced prediction score in tinnitus subjects compared to control subjects. In the second study, individuals with and without tinnitus were carefully matched for hearing levels (next to age and sex), increasing the probability that the identified differences could truly be attributed to the presence of tinnitus. Results from the first study could successfully be replicated in the second, although the effect size was notably smaller.

Throughout the manuscript, the authors provide a thoughtful interpretation of their key findings and offer several interesting directions for future studies. Their conclusions are fully supported by their findings. Moreover, the authors are sufficiently aware of the inherent limitations of cross-sectional studies.

Strengths:

The robustness of the identified differences in prediction scores between individuals with and without tinnitus is remarkable, especially as successful replication studies are rare in the tinnitus field. Moreover, the authors provide several plausible explanations for the decline of the effect size observed in the second study.

The rigorous matching for hearing loss, in addition to age and sex, in the second study is an important strength. This ensures that the identified differences cannot be attributed to differences in hearing levels between the groups.

The used methodology is explained clearly and in detail, ensuring that the used paradigms may be employed by other researchers in future studies. Moreover, the registering of the data collection and analysis methods for Study 2 as a Registered Report should be commended, as the authors have clearly adhered to the methods as registered.

---

## [Referee Report · Reviewer #2 (Public review)]

Summary:

This study aimed to test experimentally a theoretical framework that aims to explain the perception of tinnitus, i.e., the perception of a phantom sound in the absence of external stimuli, through differences in auditory predictive coding patterns. To this aim, the researchers compared the neural activity preceding and following the perception of a sound using MEG in two different studies. The sounds could be highly predictable or random, depending on the experimental condition. They revealed that individuals with tinnitus and controls had different anticipatory predictions. This finding is a major step in characterizing the top-down mechanisms underlying sound perception in individuals with tinnitus.

Strengths:

This article uses an elegant, well-constructed paradigm to assess the neural dynamics underlying auditory prediction. The findings presented in the first experiment were partially replicated in the second experiment, which included 80 participants. This large number of participants for an MEG study ensures very good statistical power and a strong level of evidence. The authors used advanced analysis techniques - Multivariate Pattern Analysis (MVPA) and classifier weights projection - to determine the neural patterns underlying the anticipation and perception of a sound for individuals with or without tinnitus. The authors evidenced different auditory prediction patterns associated with tinnitus. Overall, the conclusions of this paper are well supported, and the limitations of the study are clearly addressed and discussed.

---

## [Author Response]

The following is the authors’ response to the previous reviews.

**eLife Assessment**
This important work presents two studies on predictive processes in subjects with and without tinnitus. The evidence supporting the authors' claims is compelling, as their second study serves as an independent replication of the first. Rigorous matching between study groups was performed, especially in the second study, increasing the probability that the identified differences in predictive processing can truly be attributed to the presence of tinnitus. This work will be of interest to researchers, especially neuroscientists, in the tinnitus field.

We thank the editors at elife very much for their favorable assessment of our manuscript. Based upon the comments of the reviewer, we aimed to further improve our manuscript to be a valuable addition to the tinnitus research field.

**Public Reviews:**

**Reviewer #2 (Public review):**
Summary:This study aimed to test experimentally a theoretical framework that aims to explain the perception of tinnitus, i.e., the perception of a phantom sound in the absence of external stimuli, through differences in auditory predictive coding patterns. To this aim, the researchers compared the neural activity preceding and following the perception of a sound using MEG in two different studies. The sounds could be highly predictable or random, depending on the experimental condition. They revealed that individuals with tinnitus and controls had different anticipatory predictions. This finding is a major step in characterizing the top-down mechanisms underlying sound perception in individuals with tinnitus.Strengths:This article uses an elegant, well-constructed paradigm to assess the neural dynamics underlying auditory prediction. The findings presented in the first experiment were partially replicated in the second experiment, which included 80 participants. This large number of participants for an MEG study ensures very good statistical power and a strong level of evidence. The authors used advanced analysis techniques - Multivariate Pattern Analysis (MVPA) and classifier weights projection - to determine the neural patterns underlying the anticipation and perception of a sound for individuals with or without tinnitus. The authors evidenced different auditory prediction patterns associated with tinnitus. Overall, the conclusions of this paper are well supported, and the limitations of the study are clearly addressed and discussed.Weaknesses:Even though the authors took care of matching the participants in age and sex, the control could be more precise. Tinnitus is associated with various comorbidities, such as hearing loss, anxiety, depression, or sleep disorders. The authors assessed individuals' hearing thresholds with a pure tone audiogram, but they did not take into account the high frequencies (6 kHz to 16 kHz) in the patient/control matching. Moreover, other hearing dysfunctions, such as speech-in-noise deficits or hyperacusis, could have been taken into account to reinforce their claim that the observed predictive pattern was not linked to hearing deficits. Mental health and sleep disorders could also have been considered more precisely, as they were accounted for only indirectly with the score of the 10-item mini-TQ questionnaire evaluating tinnitus distress. Lastly, testing the links between the individuals' scores in auditory prediction and tinnitus characteristics, such as pitch, loudness, duration, and occurrence (how often it is perceived during the day), would have been highly informative.

Thank you very much for your careful evaluation of our manuscript. We agree with you that our study design has some limitations such as the assessment of higher frequencies, comorbidities, and tinnitus characteristics. In our discussion, we aimed to acknowledge these issues for future research to improve this study design and gain more insights into neural tinnitus processes.

See e.g.:

Line 946-949:

“Additionally, we rigorously controlled for hearing loss in Study 2, however, pure-tone audiometric testing was solely performed up to 8kHz and we were therefore not able to draw conclusions regarding hearing impairments in higher frequencies and their influence on the effects.”

Line 949-954:

“Moreover, we did not screen our participants for hyperacusis. This hypersensitivity to mild sounds is widely correlated with the sensation of tinnitus and underlying neural mechanisms are potentially intertwined with tinnitus processes (Schilling et al., 2023; Yukhnovich et al., 2023; Zheng, 2020). Screening for hyperacusis in future work can therefore reveal more details on participant characteristics influencing predictive processing.”

Line 955-958:

“In both studies, tinnitus distress was not correlated with the reported prediction effects. Nevertheless, tinnitus can also be characterized by other features such as its loudness, pitch or duration which were not included in the experimental assessment.”

Line 958-963:

“Additionally, we solely used a short version of the Mini-TQ (Goebel and Hiller, 1992) in Study 2, which did not allow us to relate prediction scores to subscales like sleep disturbances which potentially influence cognitive functioning and thus predictive processing. Next to sleeping disorders and distress, tinnitus is often also accompanied by psychological comorbidities such as depression or anxiety (Langguth, 2011) which are potential confounds of the results.”

Comments on revisions:Thank you for your responses. There are a few remaining points that, if addressed, could further enhance the manuscript:- While the manuscript acknowledges the limitation of not matching groups on hearing thresholds in Study 1, a deeper analysis of participants' hearing abilities and their impact on MEG results, similar to that conducted in Study 2, would be valuable. Specifically, including a linear model that considers all frequencies, group membership, and their interactions could highlight differences across groups. Additionally, examining the effect of high-frequency hearing loss on prediction scores, as performed in Study 2, would strengthen the analysis, particularly given the trend noted (line 719). Such an addition could make a significant contribution to the literature by exploring how hearing abilities may influence prediction patterns.

We appreciate your feedback and agree with you that it is a crucial question how hearing abilities influence prediction patterns in tinnitus. However, as hearing status was not assessed in the control group in study 1, we are unfortunately not able to include linear models to investigate differences across groups in this sample. This led us to the implementation of study 2 with a comprehensive hearing assessment to investigate group differences. We highlighted this issue in our methods section.

Line 170-172:

“As pure-tone audiometric testing was not included for the control subjects, group comparisons between hearing thresholds were not feasible.”

- The connection with the hippocampal regions (line 864) remains somewhat unclear. While the inclusion of the Paquette reference appropriately links temporal region activity with tinnitus, it does not fully support the statement: "An increased focus on hippocampal regions, e.g., in fMRI, patient, or animal studies, could be a worthwhile complement to our MEG work, given the outstanding relevance of medial temporal areas in the formation of associations in statistical learning paradigms"

Thank you for your constructive input. This section is purely speculative, and we do not aim to provide strong claims or expected results but solely point out potential future research directions.

- Authors should add a comparison of participants mini-TQ scores on both studies

We appreciate your input and added a comparison of mini TQ-scores between samples. For study 1, all subscales were included, however, we computed the comparison solely based on the items of the mini-TQ to increase comparability. The results were not significant, i.e., tinnitus distress values did not differ between studies.

Line 629-632:

“We additionally compared tinnitus distress values assessed by the mini-TQ (Goebel and Hiller, 1992) between study 1 and study 2 to detect potential differences between the samples, however, results of the Welch’s t-test were not significant with *t*(30.7)=1.27, *p*=.214.”

- Authors should add significant level on Fig 6.B as in Fig 3.C, and a n.s on Fig 6.D

Thank you very much for your input, we added significance levels and a n.s. to the Figures 6B and 6D.